# The GenoDiabMar Registry: A Collaborative Research Platform of Type 2 Diabetes Patients

**DOI:** 10.3390/jcm11051431

**Published:** 2022-03-05

**Authors:** Adriana Sierra, Sol Otero, Eva Rodríguez, Anna Faura, María Vera, Marta Riera, Vanesa Palau, Xavier Durán, Anna Costa-Garrido, Laia Sans, Eva Márquez, Vladimir Poposki, Josep Franch-Nadal, Xavier Mundet, Anna Oliveras, Marta Crespo, Julio Pascual, Clara Barrios

**Affiliations:** 1Department of Nephrology, Hospital del Mar, Institut Hospital del Mar d’Investigacions Mèdiques, 08003 Barcelona, Spain; asierra@psmar.cat (A.S.); erodriguezg@psmar.cat (E.R.); afaura@psmar.cat (A.F.); mvera@psmar.cat (M.V.); mriera1@imim.es (M.R.); vpalau@imim.es (V.P.); lsans@psmar.cat (L.S.); eva.marquez.mosquera@psmar.cat (E.M.); aoliveras@psmar.cat (A.O.); mcrespo@psmar.cat (M.C.); julpascual@gmail.com (J.P.); 2Department of Nephrology, Consorci Sanitari Alt Penedès-Garraf, 08800 Barcelona, Spain; sospetita@hotmail.com; 3Methodological and Biostatistical Advisory Service, Institut Hospital del Mar d’Investigacions Mèdiques, 08003 Barcelona, Spain; xduran@imim.es (X.D.); anna.costaga@e-campus.uab.cat (A.C.-G.); 4Department of Ophthalmology, Hospital del Mar, Institut Hospital del Mar d’Investigacions Mèdiques, 08003 Barcelona, Spain; vpoposki@psmar.cat; 5Research Support Unit, University Institute for Research in Primary Care, Jordi Gol (IDIAP Jordi Gol), 08041 Barcelona, Spain; josep.franch@gmail.com (J.F.-N.); xmundet.bcn.ics@gencat.cat (X.M.); 6Biomedical Research Centre in Diabetes and Associated Metabolic Disorders (CIBERDEM), 28029 Barcelona, Spain; 7Departamento de Medicina, Universidad Autonoma de Barcelona, 08193 Bellaterra, Spain

**Keywords:** type 2 diabetes, real-world patient registry, epidemiology, diabetes complication, renal function, sex differences, multiomic collaborative platform

## Abstract

The GenoDiabMar registry is a prospective study that aims to provide data on demographic, biochemical, and clinical changes in type 2 diabetic (T2D) patients attending real medical outpatient consultations. This registry is also used to find new biomarkers related to the micro- and macrovascular complications of T2D, with a particular focus on diabetic nephropathy. With this purpose, longitudinal serum and urine samples, DNA banking, and data on 227 metabolomics profiles, 77 immunoglobulin G glycomics traits, and other emerging biomarkers were recorded in this cohort. In this study, we show a detailed longitudinal description of the clinical and analytical parameters of this registry, with a special focus on the progress of renal function and cardiovascular events. The main objective is to analyze whether there are differential risk factors for renal function deterioration between sexes, as well as to analyze cardiovascular events and mortality in this population. In total, 650 patients with a median age of 69 (14) with different grades of chronic kidney disease—G1–G2 (eGFR > 90–60 mL/min/1.73 m^2^) 50.3%, G3 (eGFR; 59–30 mL/min/1.73 m^2^) 31.4%, G4 (eGFR; 29–15 mL/min/1.73 m^2^) 10.8%, and G5 (eGFR < 15 mL/min/1.73 m^2^) 7.5%—were followed up for 4.7 (0.65) years. Regardless of albuminuria, women lost 0.93 (0.40–1.46) fewer glomerular filtration units per year than men. A total of 17% of the participants experienced rapid deterioration of renal function, 75.2% of whom were men, with differential risk factors between sexes—severe macroalbuminuria > 300 mg/g for men OR [IQ] 2.40 [1.29:4.44] and concomitant peripheral vascular disease 3.32 [1.10:9.57] for women. Overall mortality of 23% was detected (38% of which was due to cardiovascular etiology). We showed that kidney function declined faster in men, with different risk factors compared to women. Patients with T2D and kidney involvement have very high mortality and an important cardiovascular burden. This cohort is proposed as a great tool for scientific collaboration for studies, whether they are focused on T2D, or whether they are interested in comparing differential markers between diabetic and non-diabetic populations.

## 1. Introduction

Diabetes mellitus is of pandemic proportions, affecting more than 450 million people worldwide, of whom up to 95% have type 2 diabetes [1]. The main complications of diabetes—both microvascular (e.g., retinopathy, neuropathy, nephropathy) and macrovascular (e.g., ischemic cardiopathy, peripheral vascular disease, cerebrovascular disease)—increase the costs associated, and entail an important reduction in the quantity and the quality of patients’ life [2,3,4]. 

In recent decades, the improvement in prevention strategies and therapeutic interventions has led to a significant reduction in most diabetes complications. However, this is not so evident in the case of diabetic kidney disease (DKD), which remains the leading cause of end-stage renal disease (ESRD) in Western countries [5,6]. When diabetes induces renal damage, patients have a higher risk of suffering from an endothelial disease in any other territory of the body, and patients with DKD have the highest cardiovascular (CV) risk and mortality among patients with chronic kidney disease (CKD) [7,8,9]. Therefore, it is essential to focus attention and effort on the early prevention, diagnosis, and treatment of DKD. Despite the high prevalence and increasing incidence of this disease, the underlying pathophysiological mechanisms are not fully understood and, even today, highly sensitive and specific diagnostic tests are not available. In this way, classical biomarkers used to estimate glomerular filtration rate (eGFR) and renal damage—such as serum creatinine and albuminuria—have well-known limitations [10,11,12,13,14], and may fail in the early detection of kidney impairment. 

In past years, high-throughput techniques have shown the feasibility of finding new biomarkers of early kidney dysfunction, as well as providing valuable information on the metabolic pathways involved in the physiopathology of DKD and other diabetic micro- and macrovascular complications [15,16,17,18,19,20,21,22,23,24]. 

The analysis of hundreds of metabolites, protein glycosylation profiles, genetic variants, or emerging biomarkers requires large-sample-size cohorts in order to robustly detect associations. GenoDiabMar is a detailed cohort useful as a tool to improve and expand knowledge on different pathophysiological pathways involved in diabetic complications, and enables the replication of results obtained in different populations to generate collaborative research. 

We present the GenoDiabMar registry, created with the aim of providing data on demographic, biochemical, and longitudinal clinical changes in a population of T2D patients attending a real medical outpatient consultation. Moreover, this registry is used to find new emerging biomarkers related to the micro- and macrovascular complications of T2D, with a particular focus on diabetic nephropathy. In this study, we analyzed the clinical and analytical characteristics at baseline and during the almost 5-year follow-up of this population. We assessed whether there were differential risk factors between the sexes in the evolution of renal function, as well as CV events and mortality associated with this population.

## 2. Materials and Methods

### 2.1. Study Design

The GenoDiabMar registry was designed as a prospective study, and has currently collected information regarding 650 Caucasian adults with T2D recruited from the nephrologist consultant of Hospital del Mar and six primary care centers from the Hospital del Mar health area, Litoral-Mar, Barcelona, Spain. The inclusion criteria were adults over 45 years old, diagnosed with T2D at least 10 years before the first study visit, if there was no renal disease—defined as glomerular filtration rate (eGFR) > 60 mL/min/1.73 m^2^ and normal urine albumin to creatinine ratio < 30 mg/g), and if renal damage was present at any time. Additional information is available in the Appendix A).

Between February 2012 and July 2015, 650 T2D patients underwent a basal in-person medical visit (V1) performed by a nephrologist and a nurse. Medical history, demographics, physical examination, and laboratory data were registered along with collection of blood and urine samples. In addition, an annual follow-up of all participants included at the baseline visit was performed in order to obtain complete analytical and clinical parameters, including new cardiovascular events and changes in the status of diabetic retinopathy and nephropathy, as well as mortality, by consulting participants´ electronic clinical reports. Between March 2017 and February 2020, living patients with functioning kidneys who did not require renal replacement therapy underwent the second in-person visit (V2). Again, analytical, and clinical data, including changes in treatments, were registered. The second biological samples for the biobank were collected on this second visit, performed an average of 4.7 (0.65) years from the baseline visit. 

### 2.2. Data Registry

#### Medical Records and CV Risk Factors Assessment

At the first visit, each participant completed a comprehensive questionnaire about their medical history, including information related to the presence and type of diabetes mellitus in their family history. A complete list of data and samples available at the baseline visit, last visit, and annually during the follow-up are summarized in Figure 1. As depicted, smoking status, history of CV events, hypertension, dyslipidemia, diabetic retinopathy (DR), and medication in use at baseline, along with changes during the follow-up, were recorded. For a detailed definition of how all of the variables were recorded, see the Appendix A. 

### 2.3. Laboratory Data and Sample Management

At baseline (V1) and at the last visits (V2), fasting venous blood and urine samples were collected. Serum, urine, DNA, and whole blood samples were stored in freezers of the Nephropathies Research Group (GREN) at the Institut Hospital del Mar d´Investigacions Mèdiques (IMIM) [25] and the Parc de Salut Mar Biobank (MARBiobanc) [26]. All samples for clinical analysis were centralized in a single laboratory—the Catalan Reference Laboratory (LRC). The main variables are summarized in Table 1 and Table 2. 

Renal function was measured as estimated glomerular filtration rate (eGFR) from calibrated serum creatinine using the Chronic Kidney Disease Epidemiology Collaboration (CKD-EPI) equation [27]. Moderate albuminuria was defined as a urine albumin-to-creatinine ratio (ACR) of 30–299 mg/g, and severe albuminuria was defined as a urine ACR of 300 mg/g or greater. DKD was defined as eGFR < 60 mL/min/1.73 m^2^ and albuminuria > 300 mg/g, or albuminuria 30–299 mg/g and DR, regardless the eGFR. Patients were classified based on their degree of kidney disease following the KDIGO guidelines as grade 1–2 if eGFR > 90–60 mL/min/1.73 m^2^, grade 3 if eGFR = 59–30 mL/min/1.73 m^2^, grade 4 if eGFR = 29–15 mL/min/1.73 m^2^, and grade 5 if eGFR < 15 mL/min/1.73 m^2^ [28].

Novel molecules and biomarkers—Alongside the conventional epidemiological phenotypes assessed by questionnaires, clinical visits, and analytical and medical reports, the GenoDiabMar registry also benefits from high-throughput techniques to assess new biomarkers related to T2D complications. 

Metabolomic profiles—Metabolic profiles of 227 metabolic traits, 143 metabolite concentrations, 80 lipid ratios, 3 lipoprotein particle sizes, and a semi-quantitative measure of albumin were determined by Nightingale Health Ltd. (Helsinki, Finland), using a targeted NMR (nuclear magnetic resonance) spectroscopy platform that has been extensively applied for biomarker profiling, as described previously [29,30,31,32].

Immunoglobulin G (IgG) glycan analysis—Variations in IgG’s glycan structures influence the effector function of IgG, modulating its immune response as proinflammatory or anti-inflammatory. IgG glycans have been associated with a high variety of conditions, including CKD [23,33]. 

All of the GenoDiabMar participants had 76 IgG glycan profiles analyzed by ultra-performance liquid chromatography (UPLC) in GENOS Glycoscience Research Laboratory (Zagreb, Croatia).

Other biomarkers available—The cohort also has information on emerging biomarkers of cardiovascular damage, measured in a targeted manner in order to study their role in kidney damage associated with T2D, as well as how they are influenced by kidney function and albuminuria. In this way, we carried out the determination of galectin 3 [34,35] and succinate [36,37,38,39] in a subset of participants. 

DNA banking—To facilitate future genetic studies, all of the participants in the registry underwent DNA extraction from the whole blood sample obtained at the baseline visit. DNA was extracted via an automated method using the QIAsymphony DSP DNA kit for whole blood, in the MARBiobanc facilities. 

## 3. Results

### 3.1. General Characteristics

The most relevant clinical characteristics and analytical variables of the cohort are displayed in Table 1 and Table 2. A total of 650 participants—61% men and 39% women, aged 69 [14] with a median time of diabetes of 15 [10] years—underwent the first visit. Of those, 356 (54.7%) had diabetic kidney disease at baseline, and the distribution per degree of chronic kidney disease was G1–G2 50.3%, G3 31.4%, G4 10.8%, and G5 7.5%. Roughly 5 years later (last in-person visit), 442 participants with functioning kidneys completed the follow-up (Table 2). As expected, the presence of DR was significantly more frequent as the glomerular filtration rate worsened, and was present in 25.8% at baseline and 30.5% at the last visit. The median body mass index (BMI) was 29.9 [6.8] kg/m^2^. Among the participants, 47.1% had a history of associated family history of T2D; 18% of the patients were active smokers, while 37.4% were ex-smokers. It should be noted that as the degree of CKD worsened, we found a significantly lower percentage of smokers. The prevalence of arterial hypertension (HBP) was high, with 91.4% of the population being affected and 77.2% having dyslipidemia. Regarding the history of previous CV events, 20.6% had a history of ischemic heart disease, with a higher prevalence peak in individuals with grade 4 CKD; 10.5% had suffered from cerebrovascular disease, and 19.8% had peripheral vascular disease—both ailments again having a higher prevalence in grade 4 CKD. With respect to antihypertensive treatment, 29.8% received angiotensin-converting enzyme inhibitors (ACEIs), 40.6% angiotensin receptor blockers (ARBs), and 2.8% a combination of ACEIs and ARBs. As detailed in the tables, as glomerular filtration rates worsen, inhibitors of the renin–angiotensin system usage decrease, with a significant drop in grades 4–5. In addition, 78% of the participants received calcium channel blockers, beta-blockers, or diuretics; 72.6% used statins, with a higher prevalence in grade 4 CKD. Concerning antidiabetic drugs, 46.3% of the participants were taking oral antidiabetic drugs (OADs), 24.3% insulin, and 28.3% both treatments combined (OADs and insulin). As detailed in the tables, the use of OADs decreases significantly as the eGFR worsens, with a clear increase in insulin usage, which was the treatment of choice in 79.6% of patients with grade 5 CKD at the baseline visit. It should be noted that the use of drugs with a demonstrated nephroprotective effect—such as sodium–glucose co-transporter inhibitors (SGLT2i) or glucagon-like peptide-1 receptor agonists (GLP1-RA)—was practically anecdotal at the baseline visit (years 2012–2015); however, we observed a clear trend towards an increase in the prescription of these drugs at the final visit (V2). However, in 2020, the percentage of patients prescribed these drugs was still kept far from the current clinical practice guidelines’ recommendations. These data should aware physicians who care for diabetic patients, and prompt them to review the existing biases between the guidelines and real clinical practice [40,41]. 

**Table 1 jcm-11-01431-t001:** General characteristics at baseline visits by grades of chronic kidney disease.

CKD Grade	1–2	3	4	5	*p*
N	327	204	70	49	
Age (years)	67 (10)	75 (13)	76 (10)	81 (22)	<0.001
Time of diabetes (years)	14 (10)	15 (10)	17 (12)	14 (4)	<0.001
Gender (Male/Female) (%)	61.8/38.2	61.6/38.2	55.7/44.3	61.3/38.8	0.812
BMI (kg/m^2^)	29.7 (6.8)	30.5 (6.6)	30.3 (7.6)	23.4 (4)	0.045
Smokers/former smokers (%)	24.8/34.5	15.2/39.2	4.3/45.7	4.1/36.7	<0.001
HBP (%)	69.4	98.5	95.7	100	<0.001
Antihypertensive treatment ACEi/ARB/ACEi + ARB %	32.7/39.7/3.1	35.3/45.6/2.9	11.4/41.4/5.7	14.3/49/0	<0.001
Cardiovascular events history (%)	31.5	46.1	60	48.9	<0.001
Ischemic cardiomyopathy	12.8	26.5	34.3	28.6	<0.001
Cerebrovascular disease	9.5	11.3	37.1	10.2	0.822
Peripheral vascular disease	14.7	23	12.9	22.5	0.019
Diabetic retinopathy (%)	17.1	27.9	34.3	63.3	<0.001
Lipid-lowering therapy (statin %)	71.8	80.9	85.7	85.7	0.002
Antidiabetic treatment (%)					
Oral agents/insulin/combined	59.3/6.1/33.9	43.1/28.4/26.9	7.3/58.6/20	8.1/79.6/6.1	<0.001
iDPP4/SGLT2i/GLP1-RA	7.6/0.3/0.9	1.9/0.5/0.5	1.4/0/0	0/0/0	0.808
eGFR (mL/min 71.73 m^2^)	82.2 (24.1)	42.8 (13.2)	23.5 (19.6)	9.14 (3.75)	<0.001
Urinary albumin/creatinine (mg/g)	9.5 (53.3)	85.8 (434)	465 (1574.7)	1158 (3210.8)	<0.001
HbA1c (mmol/mol)	60.1 (17.9)	60.6 (19.6)	59.1 (19.6)	53 (13)	0.004
Cholesterol (mg/dL)					
Total	173 (45)	165 (49)	165.5 (54)	143 (42)	<0.001
LDL	96 (36.5)	87 (36)	91 (42)	71 (35)	<0.001
HDL	45 (14)	45.2 (19)	42 (19)	43 (13)	0.665
Triglycerides (mg/dL)	129 (91.7)	144 (86)	141 (93)	125 (60)	0.072
Uric acid (mg/dL)	5.4 (1.9)	6.6 (1.7)	7 (1.9)	6 (0.8)	<0.001
Hemoglobin (mg/dL)	13.6 (2.1)	12.6 (1.88)	11.5 (1.45)	12.1 (2)	<0.001

Grade 1–2: eGFR > 90–60 mL/min/1.73 m^2^; grade 3: eGFR = 59–30 mL/min/1.73 m^2^; grade 4: eGFR = 29–15 mL/min/1.73 m^2^; grade 5: eGFR < 15 mL/min/1.73 m^2^. CKD: chronic kidney disease; BMI: body mass index; HBP: high blood pressure; ACEI/ARB: angiotensin-converting enzyme (ACE) inhibitor/angiotensin II receptor blocker; iDPP4: inhibitors of dipeptidyl peptidase 4 SGLT2i: sodium–glucose co-transporter inhibitor; GLP1-RA: glucagon-like peptide-1 receptor agonist; HbA1c: glycosylated hemoglobin; LDL: low-density lipoprotein; HDL: high-density lipoprotein. Quantitative data are expressed as medians (interquartile ranges), while qualitative variables are given in absolute and relative frequencies; *p*-values < 0.05 indicate that there are statistically significant differences between groups in the multiple comparison tests.

**Table 2 jcm-11-01431-t002:** General characteristics at baseline visits and at the end of follow-up visits.

	First Visit 2012–2015	Last Visit 2017–2020
N	650	442
Age (years)	69 (14)	72.9 (13)
Sex (male/female %)	61.1/38.9	61.5/38.8
BMI (kg/m^2^)	29.9 (6.8)	29.2 (6.2)
Family history of diabetes (%)	47.1	53.1
Cardiovascular risk factors history		
Smokers/former smoker (%)	18/37.4	15.8/41.4
High blood pressure (%)	91.4	90.9
Dyslipidemia (%)	77.2	73
Cardiovascular events history (%)	40.5	41.17
Myocardial infarction (%)	20.6	21.3
Cerebrovascular disease (%)	10.5	14.3
Peripheral vascular disease (%)	19.8	20.4
Diabetic retinopathy (%)	25.8	30.5
Antihypertensive treatment ACEI/ARB/ACEI + ARB (%)	29.8/40.6/2.8	31/37.8/2.3
Others (Calcium antagonists/ß-blockers/diuretics) (%)	78	80.8
Lipid-lowering therapy (%)		
Statins	72.6	68.1
Fibrates	10.1	6.8
Other	3.4	7.8
Antidiabetic treatment		
Oral agents only (%)	46.3	41.2
DPP4i/SGLT2i/GLP1-RA (%)	6.1/0.3/0.6	21/5.9/4.3
Insulin only (%)	24.3	21
Oral agents + insulin (%)	28.3	34.6
Diet (%)	1.1	1.1
Serum creatinine (mg/dL)	1.12 (0.81)	1.11 (0.78)
eGFR (mL/min/1.73mt2)	60.4 (46.5)	57.7 (46.4)
Urinary albumin/creatinine (mg/g)	34.2 (217.05)	32.6 (219.9)
Hemoglobin (gr/dL)	13 (2.2)	13.1 (2.5)
HbA1c (mmol/mol)	59.6 (18.5)	55.2 (18.5)
Uric acid (mg/dL)	6.1 (2.1)	6.6 (2.5)
Total cholesterol (mg/dL)	171 (48)	162 (57)
LDL cholesterol (mg/dL)	94 (39)	89 (44)
HDL cholesterol (mg/dL)	45 (16)	46 (17)
Triglycerides (mg/dL)	136 (90)	137 (90.5)

BMI: body mass index; ACEI/ARB: angiotensin-converting enzyme (ACE) inhibitors/angiotensin II receptor blockers; iDPP4: inhibitors of dipeptidyl peptidase 4; SGLT2i: sodium–glucose co-transporter inhibitors; GLP1-RA: glucagon-like peptide-1 receptor agonists; eGFR: estimated glomerular filtration rate; HbA1c: glycosylated hemoglobin; LDL: low-density lipoprotein; HDL: high-density lipoprotein. The median time between the first in-person visit and the last was 4.7 (0.65) years. Quantitative data are expressed as medians [interquartile ranges], while qualitative variables are given in absolute and relative frequencies. Losses due to mortality between the two visits were 137 patients.

### 3.2. Patient Follow-Up: Renal Deterioration by Sex and Mortality

To attain a longitudinal view of the proceeding of the main variables of the cohort, we assessed the evolution of renal function in the 611 participants with functioning kidneys at the baseline visit (non-dialysis patients). The overall mean annual glomerular filtration loss was −1.2 (−1.8:−0.5 mL/min/1.73 m^2^), being different between men and women. Men had a median eGFR higher than women at baseline (66.3 [41.8:84.3] vs. 63.4 [41.7:86.4] mL/min/1.73 m^2^), but the rate of loss of kidney function was significantly lower in women, losing 0.93 (0.40–1.46) fewer glomerular filtration units per year than men, regardless of albuminuria (Figure 2). Roughly 17% of the patients experienced rapid deterioration of renal function—defined as a loss of ≥5 mL/min/1.73 m^2^/year [28] over the follow-up period—of whom 75.2% were men and 24.7% were women. 

The parameters influencing a faster deterioration of renal function were different between the sexes. As depicted in Table 3, the presence of peripheral vascular disease was a risk factor for women, but not for men. Additionally, macroalbuminuria was a significant factor for men, but not for women. The area under the ROC curve (AUC) for predicting rapid progression based on albuminuria was 0.62 [0.55:0.68] with a cutoff value of 451 mg/g for men, and 0.70 [0.62–0.78] with a cutoff value of 18.4 mg/g for women. These findings can help to identify early clinical and analytical risk factors for worse renal evolution in a differential and more personalized manner.

Multivariate logistic regression model showing the variables of risk of undergoing a rapid decline in renal function, defined as a loss of >5 ml/min/m^2^ in estimated glomerular filtration rate (eGFR) per year, separated by sex.

Although this registry was initially started with the objective of studying new biomarkers of kidney injury in T2D, it is a well-characterized population that has detailed information on other micro- and macrovascular diabetic complications, which we also analyzed. Thus, during the follow-up period, 135 patients (22.1%) endured one or more CV events, of whom 33.3% fulfilled the established criteria for DKD and 18.4% did not match the DKD criteria at baseline (*p* < 0.001). Figure 3A shows the schematic distribution of the new onset of CV events and DR. During the follow-up, 137 patients (22.4%) developed DR or worsened their previously established DR. In addition, 75 (12.4%), 54 (8.4%), and 28 (4.38%) patients developed or worsened ischemic cardiopathy, peripheral vascular disease, and cerebrovascular disease, respectively. A higher incidence of events was observed in the group of patients who met the criteria for established DKD at the baseline visit (Figure 3B). 

There was overall mortality of 23%—38% of which was due to cardiovascular causes, and 16% due to cancer. Furthermore, throughout the follow-up, 22 patients (3.6%) started renal replacement therapy, and 10 were lost to follow-up (Figure 4 and Appendix A: Flow-diagram of patients recruitment and follow up).

## 4. Discussion

This is a well-characterized registry with information on longitudinal micro- and macrovascular complications of T2D, as well as detailed clinical and analytical information. We have provided a detailed picture of the clinical and analytical behavior of these patients, so as to facilitate detailed knowledge of the available variables, as well as their distribution and evolution. In this way, the data of this registry are closer to “real-world data” (RWD), with patients who sometimes may be excluded from other types of clinical trials or studies, with strict inclusion criteria. This allows us to obtain evidence that is closer to routine clinical practice.

The characteristics of the evolution of renal function and the differences found between the sexes in morbimortality are similar to those described in the literature [42,43,44] for DKD patients. This observation reinforces the value of this registry as a population, with renal characteristics comparable to other cohorts, so as to be able to carry out collaborative studies. It is worth noting that the risk factors of rapid deterioration of renal function, which differ between sexes, are clinical and analytical factors available in any medical consultation. This is of special value in evaluating the patient’s risk in an individualized manner during routine consultation. 

The absolute value of macrovascular events is not high compared to other large-population cohorts. However, since this cohort has patients with a high burden of disease, recruited from real consultations—which allows us a more reliable and closer follow-up of the clinical events—the relative values and percentages of events and complications are similar to or even higher than those observed in other, larger cohorts. As stated, it is important to join efforts in conducting collaborative studies to minimize statistical power problems and, although a single registry may not have enough events, management through meta-analysis of results and validation of data in external cohorts is postulated as a mandatory scientific practice. In this study, we have shown that diabetic patients with kidney disease, beyond classic antihypertensive or lipid-lowering treatments and acceptable metabolic control, have a very high mortality—mainly associated with cardiovascular events. The physicians attending this kind of patient—general practitioners, endocrinologists, cardiologists, or nephrologists—must be especially aware and approach these patients in a multidisciplinary manner. The low percentage of patients in this cohort under treatment with drugs that have demonstrated cardio- and nephroprotection (SGLT2i or GLP1-AR) is remarkable. Although we know that this trend is changing, we must continue efforts to implement these drugs in routine clinical practice. 

In addition, the main strengths of this cohort are that it includes a collection of baseline and longitudinal follow-up biological samples, and covers the entire spectrum of kidney disease, including patients from grade 1 CKD to patients on renal replacement therapy. As these are patients from medical consultations belonging to our healthcare area, we have detailed and precise information about them, and we can ensure good monitoring capacity. 

One of the limitations of the study is the generalizability of the findings, which is limited to Caucasian subjects. The results of our study may be validated in multiethnic cohorts in order to evaluate their applicability in broader populations with T2D. We do not have urine samples from all of the participants, and lack information regarding diet and lifestyle. Moreover, we have not conducted kidney histological studies in most of the patients to ensure the renal disease etiology. Since no other reliable and non-invasive markers have been established, we cannot overcome this limitation but, in order to minimize misdiagnosis, medical history—including ultrasonography and fundoscopy studies—was reviewed by two nephrologists. In addition, to date, the registry has only ensured the extraction of DNA from the samples, but they have not yet been sequenced. With current technology, the statistical power of the cohort would be insufficient to propose GWAS-type studies without a prior hypothesis. However, once the population has been sequenced, studies aimed at validating predetermined genetic variants could be considered, as well as potential replication studies.

We presented a registry of patients from real nephrology and primary care medical outpatient consultations, from which we have registered several clinical and analytical variables for 5 years. In addition, the registry includes serum and urine biobanks, DNA banks, and data on metabolomics, glycomics, and other biomarkers already analyzed. We consider this cohort to be a great potential tool for scientific collaboration for studies, whether they are focused on T2D, or whether they are interested in comparing differential markers between diabetic and non-diabetic populations. Furthermore, as we have shown in other collaborative projects, the GenoDiabMar registry can meet the criteria to replicate or meta-analyze results obtained in other cohorts. It should be noted that this registry is part of The Consortium of Metabolomics Studies (COMETS), whose main objective is to create a collaborative network to identify metabolomic markers associated with different phenotypes and pathologies [45]. 

This descriptive publication of our GenoDiabMar registry should engage researchers in collaborative efforts to advance knowledge of the etiology, diagnosis, treatment, and prognosis of T2D complications. In this spirit, we invite researchers—including those without data of their own—to join us with scientific collaboration proposals.

## Figures and Tables

**Figure 1 jcm-11-01431-f001:**
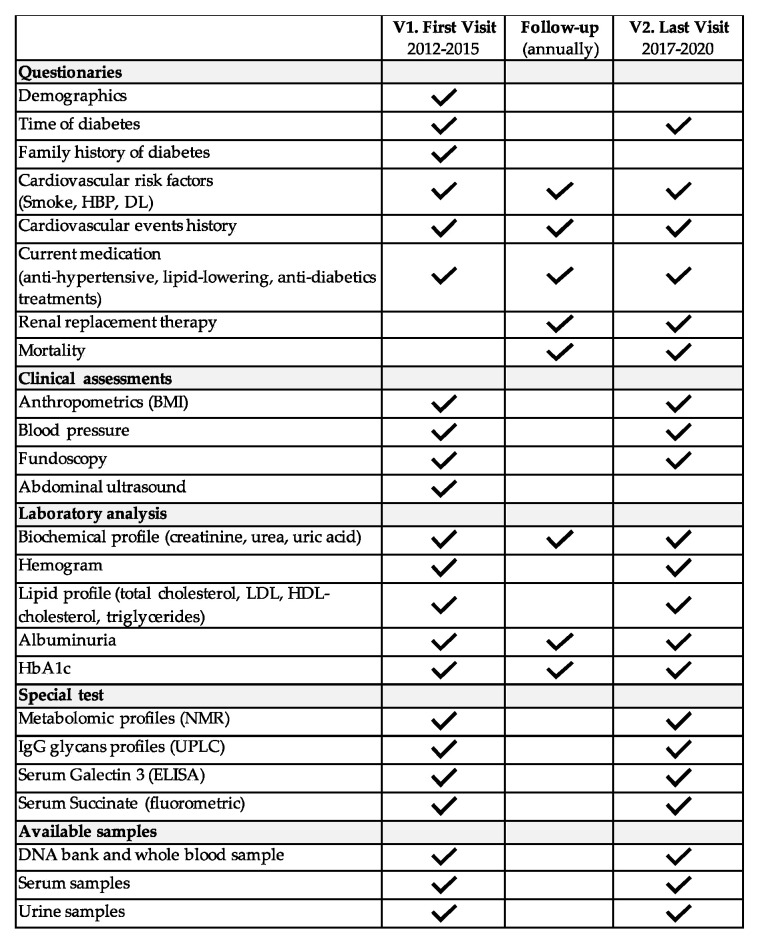
Summary of available data and samples from different visits and follow-ups: V1: first in-person visit; V2: last in-person visit; HBP: high blood pressure; DL: dyslipidemia; HbA1c: glycosylated hemoglobin; LDL: low-density lipoprotein; HDL: high-density lipoprotein; NMR: nuclear magnetic resonance spectroscopy; UPLC: ultra-performance liquid chromatography.

**Figure 2 jcm-11-01431-f002:**
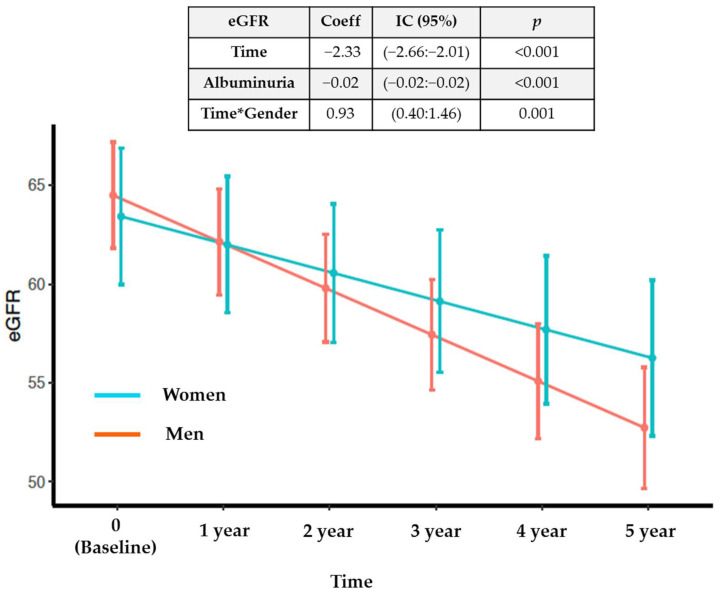
Evolution of estimated glomerular filtration rate (eGFR) by sex, adjusted by albuminuria, Time and Gender interaction (Time*Gender) in the mixed linear model, and its graphical representation of the mean eGFR between sexes.

**Figure 3 jcm-11-01431-f003:**
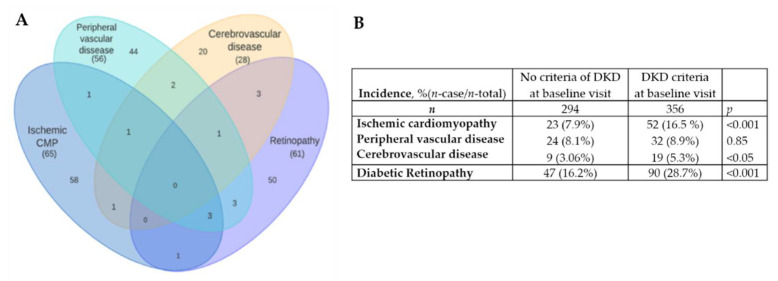
(**A**) Schematic distribution of the recorded cardiovascular events and new onset of diabetic retinopathy, throughout the follow-up period; (**B**) Incidence of the recorded cardiovascular events and diabetic retinopathy during the follow-up, by criteria of DKD at baseline visit.

**Figure 4 jcm-11-01431-f004:**
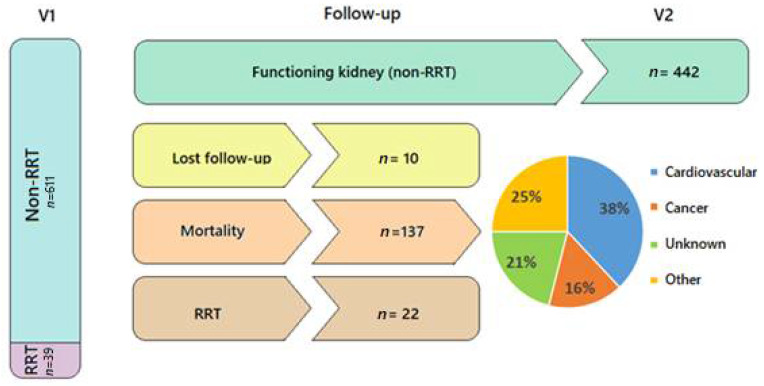
Flowchart depicting patients’ distribution during follow-up, their need for renal replacement therapy (RRT), mortality, and its causes. Pie chart shows causes of mortality distributed in percentages.

**Table 3 jcm-11-01431-t003:** Risk factors for rapid deterioration of kidney function by sex.

	Men		Women	
	OR (IC95%)	*p*	OR (IC95%)	*p*
Age	1.01 (0.98:1.04)	0.62	1.02 (0.97:1.08)	0.48
Diabetic retinopathy	1.18 (0.61:2.23)	0.61	0.99 (0.25:3.33)	0.98
Time of DM2	1.02 (0.99:1.06)	0.18	0.97 (0.91:1.03)	0.42
BMI	1.03 (0.96:1.10)	0.45	0.97 (0.89:1.05)	0.47
Ischemic cardiopathy	1.02 (0.53:1.88)	0.94	1.16 (0.31:3.39)	0.80
Peripheral vascular disease	0.79 (0.39:1.53)	0.49	3.32 (1.10:9.57)	0.02
Stroke	1.83 (0.85:3.74)	0.12	1.82 (0.38:6.21)	0.41
Albuminuria > 300 mg/g	2.40 (1.29:4.44)	0.005	0.99 (0.91:3.73)	0.99
HbA1c	0.89 (0.71:1.11)	0.32	1.14 (0.80:1.59)	0.43
Smoker	1.03 (0.46:2.30)	0.94	1.15 (0.21:4.97)	0.86
Former smoker	0.72 (0.37:1.46)	0.35	0.29 (0.02:1.62)	0.25

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
