# Peer review of "The GenoDiabMar Registry: A Collaborative Research Platform of Type 2 Diabetes Patients"

_jcm, 2022, doi:10.3390/jcm11051431_

Round 1

Reviewer 1 Report

The authors describe a T2D cohort with 650 patients having either long lasting T2D or T2D with renal complications. Patients was recruited in the Barcelona area, Spain from one hospital outpatient clinic and six primary centers around the hospital. Despite the cohort being relative small, many clinical variables are measured and blood samples are analyzed for an extensive number of biomarkers.

Generally, I find the paper relevant but way to long and unfocused. Especially the method and result section needs focusing and shortening. I would recommend the authors to consider this paper more as a cohort profile paper and leave out many of the data not relevant to describe the cohort profile. This

I would recommend to present all data as median and IQR.

General comment

  • The study recruit patients with or without renal problem, thus a focus on nephropathy is obvious, but why also include CVD (micro and macro) where n on outcome is low. This is likely to give future power-problem when looking for effects related to the different CVD outcomes?
  • I need a clear definition of the renal function in the inclusion criteria.
  • Please include a flow diagram in the method section on patient recruitment and follow-up
  • Are additional follow-up data possible and if yes then how?
  • How many patients was lost during the follow-up period (include in flow-diagram)?
  • No genetic analysis have been conducted in the cohort and one could speculate if the cohort is to small to be used for this purpose. What is the authors opinion on this?
  • Give HbA1c in mmol/mol
  • In table 2 please provide the period between V1 and V2 if this was not a fixed time pleas give as median and IQR
  • Figure 1 the time need to be absolute time not visit 1-6.
  • In my opinion, data in figure 2 are more clearly presented in a table or in plain text.
  • Move figure 4 to methods

Author Response

Review 1:

The authors describe a T2D cohort with 650 patients having either long lasting T2D or T2D with renal complications. Patients was recruited in the Barcelona area, Spain from one hospital outpatient clinic and six primary centers around the hospital. Despite the cohort being relative small, many clinical variables are measured and blood samples are analyzed for an extensive number of biomarkers.

General comment

Generally, I find the paper relevant but way to long and unfocused. Especially the method and result section needs focusing and shortening. I would recommend the authors to consider this paper more as a cohort profile paper and leave out many of the data not relevant to describe the cohort profile.

We thank the reviewer for their feedback and recommendations to improve the manuscript.

In our experience when collaborating with other groups and their cohorts we have had to give very specific details about the collection of data and biological samples, that is why we wanted to include so much detail in this manuscript. However, following your requested we have substantially shortened the methodological details in the main manuscript and created a supplementary section for complete methods.

We have tried our best to follow all the recommendations that we detail below

  • I would recommend to present all data as median and IQR.

Quantitative data were expressed as means values ± (standard deviation) for normally distributed variables and as medians [with interquartile ranges] for non-normally distributed variables, while qualitative variables are given in absolute and relative frequencies. Following your recommendation we have revised and changed the format to present the quantitative data as median and [IQR] through the text and within the tables and legends.

  • The study recruit patients with or without renal problem, thus a focus on nephropathy is obvious, but why also include CVD (micro and macro) where n on outcome is low. This is likely to give future power-problem when looking for effects related to the different CVD outcomes?

Cardiovascular disease (CVD) is the leading cause of morbidity and mortality among individuals with type 2 diabetes mellitus (T2DM), mainly if DKD is present. We present a well characterized registry that gathered longitudinal micro and macrovascular complications of T2D as well as detailed clinical and analytical information.

The absolute value of macrovascular events is not very high compared to other large population cohorts, but as they are patients with a high burden of disease, recruited from real consultations, which allows us a more reliable and close follow-up of the clinical events, lead to the relative value or the percentage of events and complications are similar and even higher than that of observed in other larger cohorts with which we collaborate.

As stated in the manuscript, it is important to join efforts in conducting collaborative studies, precisely to minimize statistical power problems and, although a single registry may not have enough events, management through meta-analysis of results and validation of data in external cohorts is postulated as a mandatory scientific practice and precisely in this way we have worked in past and current projects.

We have highlighted this idea in the discussion/conclusion section.

  • I need a clear definition of the renal function in the inclusion criteria.

The measurement of renal function and its stratification of both glomerular filtration rate and albuminuria are explained in more detail in the methods section; 2.3. Laboratory Data. Following the reviewer recommendations, we have also included the definition of normal kidney function for this population in previous section 2.1. Study Design, as follow:

“ The inclusion criteria were adults over 45 years old, diagnosed with T2D at least 10 years before the first study visit, if there was no renal disease defined as glomerular filtration rate (eGFR) >60 mL/min/1.73m2 and normal urine albumin to creatinine ratio <30 mg/g), and at any time, if renal damage was present.”

  • Please include a flow diagram in the method section on patient recruitment and follow-up

More than 1600 adult patients from our close medical environment with the diagnostic label of diabetes, were thoroughly reviewed, including medical history, ophthalmological data and ultrasound. After contacting by calls and letters with those we consider could enter the registry, 650 patients finally attended the first visit and signed the informed consent. This review was carried out in 2012 after approval by the ethics committee. Unfortunately, we have no way of knowing the reasons for screening carried out in this first review, so that we cannot detail the exact number of patients who did not meet the inclusion criteria and their reasons, the exact number of patients that we decided not to include due to very age advanced, dependence or severe comorbidities or the exact number of negatives.

Our main objective was to include the maximum number of patients with different degrees of renal involvement and with the possibility of reaching a minimum of follow-up consultations and obtaining a representative sample of type 2 diabetic patients from our consultations for whom we could have biobank and reliable clinical and analytical data.

We have already include a Flow chart figure as requested, as supplementary methods material (Current Supplementary Figure 1).

  • How many patients was lost during the follow-up period (include in flow-diagram)?

611 patients who were not undergoing renal replacement therapy in the first visit were followed up, a total of 169 were lost during the follow-up, and 442 with a functioning kidney accomplished the last in-person visit.

The current figure 4 shows the data on the loss of patients during follow-up and its causes, with special detail on the causes of death. In addition, and as required by the reviewer, we have included the loss-data during the follow-up in the flow-diagram of the supplementary material (Figure 1S).

  • Are additional follow-up data possible and if yes then how?

The follow-up of the patients ended in February 2020, however, since they are patients from our own health area from whom we have direct access, we could consult and log in their clinical and analytical variables if a study that required it were proposed. In fact this is one of the advantages of real-world cohorts

We have added a sentence in the Supplementary Material; Methods; Study design section that clarifies it as follow: “

“The follow-up of the patients ended in February 2020, however, since they are patients from our own health area from whom we have direct access, we could consult and log in their clinical and analytical variables if a study that required it were proposed. Contact can be made directly by email addressed to principal Investigator of the project, Dr. Clara Barrios or through the website of our research group on nephropathies of the Mar Institute of Medical Research (IMIM)”.

  • No genetic analysis have been conducted in the cohort and one could speculate if the cohort is to small to be used for this purpose. What is the authors opinion on this?

We thank the reviewer for this appreciation. Indeed, to date we have only ensured the extraction of DNA from the samples, but they have not yet been sequenced. With current technology, the statistical power of the cohort would be insufficient to propose GWAS-type studies without a prior hypothesis. However, once the population has been sequenced, studies aimed at validating certain genetic variants could be considered, as well as potential replication studies. We have highlighted this limitation in the discussion section.

  • Give HbA1c in mmol/mol

NGSP HbA1c (%) variable is already expressed in IFCC mmol/mol unit, as requested

  • In table 2 please provide the period between V1 and V2 if this was not a fixed time please give as median and IQR.

The first visits started consecutively February 2012 to July 2015 and the second in-person visits were from March 2017 to February 2020, after an average of 4.96 (± 0.43) years from the baseline visit. We have include this data in table 2 legend as a median [IQR] 4.7 [0.65] years, as requested.

  • Figure 1 the time need to be absolute time not visit 1-6.

We thank the reviewer for this appreciation. We have changed the concept on the time axis that currently represents the years and not the visits. We will adjust the figure to the characteristics required by the editor in case we achieve the final editing process, with this information clarified

  • In my opinion, data in figure 2 are more clearly presented in a table or in plain text.

With the purpose of representing the combination of more than one event in the same patients, we made this schematic combination of Ven. If for the reviewer it is not mandatory to withdraw it, we would prefer to leave it. Nevertheless, following your recommendations and preferences, we have extend the explanation thought the text and we have created a new figure 2 (current figure 3) with panel A and B. See the sentences and the new figure in the results section as follow:

“Thus, during the follow-up period 135 patients (22.1%) endured one or more CV events, of which 33.3% fulfilled the established criteria for DKD and 18.4% did not match the DKD criteria at baseline (p<0.001).  Figure 3 panel A shows the schematic distribution of the new onset of CV events and DR. During the follow-up, 137 patients (22.4%) developed DR or worsening of the previous stablished one. In addition, 75 (12.4%), 54 (8.4%) and 28 (4.38%) patients, developed or worsening ischemic cardiopathy, peripheral vascular disease and cerebrovascular disease, respectively. A higher incidence of events were observed in the group of patients who met the criteria for established DKD at the baseline visit (Figure 3, panel B).”

  • Move figure 4 to methods

Follow your recommendations we have moved figure 4 (current figure 1) to methods section.

Reviewer 2 Report

In this article authors explain a prospective study cohort GenoDiabMar registry that  provide data on demographic, biochemical and clinical changes, from a “real-world” population of Type 2 DM (T2D) patients.

This is a simple prospective cohort but the author want to project a new concept for the real-world” population of Type 2 DM.

Please mention what is an advantage of this type of cohort compare to other cohorts in T2D complications studies.

What is new in this article? 

Author Response

Review 2:

Comments and Suggestions for Authors

In this article authors explain a prospective study cohort GenoDiabMar registry that  provide data on demographic, biochemical and clinical changes, from a “real-world” population of Type 2 DM (T2D) patients.

This is a simple prospective cohort but the author want to project a new concept for the real-world” population of Type 2 DM.

Please mention what is an advantage of this type of cohort compare to other cohorts in T2D complications studies.

What is new in this article? 

We appreciate the reviewer for their feedback and interest.

The GenoDiabMar registry has collaborated and collaborates with different studies of metabolomics, glycomics and biomarkers, associated with kidney disease and CV risk. To date, a detailed description of the cohort, as well as the variables and available samples, had not been made. In our experience, collaborating with other groups, we have had to provide very specific details about the collection of the data and biological samples, that is why we wanted to include so much details in the manuscript. We have now substantially shorter the methods section and included a supplementary material for expanded explanations. Regarding the results we consider it of a high interest for nephrologist, cardiologist and endocrinologist since they provide a detailed picture of the clinical and analytical behavior of the population, as well as the variables available to carry out collaborative studies.

Patients included in randomized clinical trials or in large public access databases with partial information might not be representative of those seen in clinical practice. In this way, the observed conclusions may not be applicable to real clinical practice. (p.e.: doi: 10.1186/s12933-021-01323-5, https://doi.org/10.1155/2019/2018374, doi: 10.3389/fphar.2021.700012).

The knowledge and close follow-up of the participants included in records of real medical consultations helps us overcome the barriers of strict inclusion criteria, allows us to review and expand variables of interest, as well as follow-up times.

In this way, well-characterized patient registries with real world data and with biobank samples are postulated as an increasingly necessary tool to more reliably evaluate both clinical and pharmacological aspects, as well as the study of associated biomarkers.

We have included changes to the abstract, introduction, and discussion/conclusion sections to make these concepts clearer.

We thank the reviewer again for the effort and interest in contributing for improvement to the manuscript. Please find attached the document with the  changes performed, following also the recommendations of the co-reviewer.

Round 2

Reviewer 1 Report

General comments

This paper describes a database for deep pheno- and genotyping T2D patients with kidney diseases. In general, the paper is still way to long. It needs focus, structure and readability.

In the literature, the term ”real world” refers everyday life, but the data presented in this register are all clinical data (no data on lifestyle and quality of life). Thus, the term “real world” is not suited to describe this cohort. Please remove it for title and throughout the text.

The term “registry” gives the impression that this paper describe all the official patient records in the area. But this is not the case. Please replace “registry” with “database” thru the paper.

The abstract must be improved with a definition of the different G1-G5, a study aim of the paper, aim for database and a conclusion.

Specific commets

P1.l37                 is data given as mean±SD or median with IQR? I would prefer median

P1.l39                 same issue as P1.l37

P2.l55                 the sentence make no sense. Please rephrase.

P3.l84                 is the database build with the aim to collect blood samples? Think the aim need more focus.

P5.l144               unclear use of intervals; 30-299 vs 300 and greater.

P5.l158-165      section fits better to results.

P6.l183-191      section fits better in introduction and results.

Table 1               please provide full names for all abbreviations in text, state if data is given as mean or median, explain p values. Are there any missing numbers in table 1?

P8.l250-257      this section is hard to follow.

Tabel 2              how many have died fro first to last visit?

                             Time of diabetes is irrelevant in this table

P10.l273            why only 611 when you have 650?

Figure 4             nice figure but I do not understand the pie chart.

P13.l347            All patients are hopefully from “the real world” J.

P8.l366-372      the cohort is likely to small for genetic studies

P14.l382-385    has data been published elsewhere?

Author Response

Comments and Suggestions for Authors

We appreciate the reviewers for the comments, their feedback and recommendations to improve the manuscript. We have tried our best to improve the manuscript according to your comments as well as our own consideration.

General comments

  • This paper describes a database for deep pheno- and genotyping T2D patients with kidney diseases. In general, the paper is still way to long. It needs focus, structure and readability.

Following former requests we have substantially shortened the methodological details in the main manuscript and created a supplementary section for complete methods.

Again, and following your current appreciation, we have minimized the amount of information that we had included in previous versions of the manuscript related to the description of IgG glycosylation, metabolomic data and other emerging biomarkers that the registry has available. We understand the reviewer's point of view that these details may deviate the focus and the current descriptive value of the manuscript.

We also consider important to point out that the registry account with data on metabolomics, glycomics or emerging biomarkers available. As we discussed in the manuscript, in past years, high-throughput techniques have shown the feasibility of finding new biomarkers of early kidney dysfunction, as well as providing valuable information on the metabolic pathways involved in the physiopathology of DKD and other diabetic micro and macrovascular complications. This kind of analysis requires large sample size cohorts to robustly detect associations. In that way, collaborative studies are essential and the GenoDiabMar study is a detailed cohort useful as a tool to improve and expand knowledge on different pathophysiological pathways involved in diabetic complications and allow to replicate results obtained in different populations to generate collaborative research.

In addition, we have rewritten and restructured the abstract and the main text, to make the objectives of this study clearer. The objectives are encompassed in two main aspects. On the one hand, we carry out a detailed description and assessment of the clinical and analytical variables of a population with type 2 diabetes in our health care area, with special attention to the evolution of renal function in general and separated by sex and the evolution of cardiovascular risk factors. On the other hand, we want to make known the variables, biological samples and multiomic determinations that the registry has to engage researchers in collaborative efforts to advance knowledge of the etiology, diagnosis, treatment, and prognosis of T2D complications.

  • In the literature, the term ”real world” refers everyday life, but the data presented in this register are all clinical data (no data on lifestyle and quality of life). Thus, the term “real world” is not suited to describe this cohort. Please remove it for title and throughout the text.

Patients included in randomized clinical trials or in large public access databases with partial information might not be representative of those seen in clinical practice. In this way, the observed conclusions may not be applicable to real clinical practice. (p.e.: doi: 10.1186/s12933-021-01323-5, https://doi.org/10.1155/2019/2018374, doi: 10.3389/fphar.2021.700012).

The knowledge and close follow-up of the participants included in records of in person medical consultations helps us overcome the barriers of strict inclusion criteria, allows us to review and expand variables of interest, as well as follow-up times.

In this way, well-characterized patient registries with real world data (RWD) and with biobank samples are postulated as an increasingly necessary tool to more reliably evaluate both clinical and pharmacological aspects, as well as the study of associated biomarkers.

Real-world data (RWD) are data that come from sources other than traditional clinical trials and are becoming increasingly important to today’s healthcare decisions

The FDA for example define the term RWD as "data relating to patient health status and/or the delivery of health care routinely collected from a variety of sources. Examples of RWD include data derived from electronic health records, medical claims and billing data; data from product and disease registries; patient-generated data, including from in-home-use settings; and data gathered from other sources that can inform on health status, such as mobile devices".

However, it does not specify that lifestyle or quality of life aspects should mandatory be included. In any case, if it is not clear to the reviewer that this term can be used in this registry, we will follow their valuable recommendations.

Notice we have previously included changes to the abstract, introduction, and discussion/conclusion sections to make these concepts clearer in the previous modified version of the manuscript. Now, and follow the reviewers suggestion we have withdrawn the term from the title and we have nuanced the concept through the text.

  • The term “registry” gives the impression that this paper describe all the official patient records in the area. But this is not the case. Please replace “registry” with “database” thru the paper.

Patient registries have been defined as “an organized system that uses observational study methods to collect uniform data (clinical and others) to evaluate specified outcomes for a population defined by a particular disease, condition, or exposure, and that serves a predetermined scientific, clinical, or policy purpose(s).” p.e. PMID: 21204321. “A patient registry is a collection—for one or more purposes—of standardized information about a group of patients who share a condition or experience

GenoDiabMar is a registry with longitudinal data of patients who share a common condition (Type 2 Diabetes) included through an organized system. It meets the definition of registry, which does not specify that all the official patient records in the area must be included. This project was thus recognized by the local and National institutions that have financed it and we ask the reviewer to please reconsider this suggestion to modify the registry concept by dataset.

  • The abstract must be improved with a definition of the different G1-G5, a study aim of the paper, aim for database and a conclusion.

We have currently modified the abstract and the main text as we said in the general comments

Specific commets:

P1.l37                 is data given as mean±SD or median with IQR? I would prefer median

P1.l39                 same issue as P1.l37

Yes, following your former suggestions and as we said in our previous rebuttal, we present quantitative data as median and [IQR] through the text (including abstract) and tables.  

P2.l55                 the sentence make no sense. Please rephrase.

Thanks for the appreciation. We have rephrased the sentence

P3.l84                 is the database build with the aim to collect blood samples? Think the aim need more focus.

We have currently modified the abstract and the main text as we said in the general comment.

P5.l144               unclear use of intervals; 30-299 vs 300 and greater.

Diabetes kidney disease has a clearer definition when there is severe albuminuria (>300mg/g or grade A3) or a sustained decrease in glomerular filtration <60ml/min, a grade that already defines the presence of chronic kidney disease. However, when there is moderate albuminuria (30-299mg/g or A2), and a glomerular filtration rate >60, it is not so clear whether or not the patient has diabetes associated renal damage, that is why, to minimize classification biases, this group of patients is included in the DKD group if they associate the presence of diabetic retinopathy. This is so because when diabetes produce microvascular affection of retinal the majority (75-80%) of patients also associate diabetic kidney damage (KDIGO guidelines)

P5.l158-165      section fits better to results.

P6.l183-191      section fits better in introduction and results.

As we said in general comments, we have substantially reduce this section which is currently maintained as a basic description of the data on metabolomics, glycomics and other markers that are available in the registry

Table 1               Please provide full names for all abbreviations in text, state if data is given as mean or median, explain p values. Are there any missing numbers in table 1?

We thanks the reviewer for this appreciation. The legend of the tables currently explains the full name of the abbreviations, in addition and following your suggestions, we have added these full names when they appeared throughout the text. As legends shown, quantitative data are expressed as medians [interquartile ranges], while qualitative variables are given in absolute and relative frequencies. We have currently included the p value meaning as well. No, there are no missing data, the value “0” means there were no patients under these conditions.

P8.l250-257      This section is hard to follow.

Lines 250-257 said: “Of note, the use of drugs with a demonstrated nephroprotective effect such as Sodium glucose co-transporter inhibitors (SGLT2i) or Glucagon-like peptide-1 receptor agonists (GLP1-RA) at the baseline visit (years 2012-2015) was practically anecdotal, with a clear tendency to increase their prescription on the final visit. Albeit, at the beginning of 2020, they were still far from being part of the treatment in most of these patients and, in line with other studies, the actual use of these drugs kept apart from the current clinical practice guidelines (54-55).”

We thank the reviewer for this comment to improve this section. In recent years and as a result of the conclusions of large clinical trials (p.e. CREDENCE, DAPA-CKD, REWIND, FLOW, SOUL among others) that have shown renal and cardiovascular benefit of these two pharmacological groups, the guidelines for the management of diabetic patients have changed radically.

In that way, it recommend prioritizing the use of these molecules when patients have kidney damage. However, the practical implementation of these recommendations is still far from being reflected in clinical practice. This observation with real data from our health area is very interesting and can be extrapolated to clinical practice when observed with pharmacological prescription data in broader health areas, for example the area of Catalonia (CatSalut). We want to highlight the existing step between the recommendations of the guidelines and the clinical practice with our data closer to the RWD. We have rephrased the paragraph to improve and clarify the idea we want to convey.

Table 2              How many have died from first to last visit? Time of diabetes is irrelevant in this table.

In the following section entitled: “3.2. Patients follow up. Renal progression by sex and mortality”, we specified the percentage of losses due to death observed during follow-up as: “There was overall mortality of 23%, 38% due to cardiovascular causes, and 16% due to cancer. Also, throughout the follow-up, 22 patients (3.6%) started renal replacement therapy and 10 were lost to follow-up (Figure 4 and flow-chart as supplementary material). Thus, as figure 4 and Supl.Figure 1 depicts the overall mortality was 137 patients. We have now include this data in table 2 legend. Also, we have remove “Time of diabetes” as the reviewer suggested

P10.l273            why only 611 when you have 650?

Manuscript sentence “To have a longitudinal view of the proceeding of the main variables of the cohort, we assessed the evolution of renal function in the 611 participants with functioning kidneys at the baseline visit (non-dialysis patients)”. See also figure 4.

It is not possible to do longitudinal follow-up of renal function in patients undergoing renal replacement therapy (RRT), in general patients undergoing hemodialysis or kidney transplantation, since they do not have their own functioning kidneys to assess their evolution.

Figure 4             Nice figure but I do not understand the pie chart.

We appreciate your comment. The Pie-Chart shows the distribution in percentages of the different causes of mortality. To clarify pie-chart meaning we have now include a sentence in the legend

P13.l347            All patients are hopefully from “the real world” J.

P8.l366-372      the cohort is likely to small for genetic studies

As we said in our previous rebuttal, we agree with the reviewer. Indeed, to date we have only ensured the extraction of DNA from the samples, but they have not yet been sequenced. With current technology, the statistical power of the cohort would be insufficient to propose GWAS-type studies without a prior hypothesis. However, once the population has been sequenced, studies aimed at validating certain genetic variants could be considered, as well as potential replication studies. We revised and highlighted this limitation in the discussion section.

P14.l382-385    has data been published elsewhere?

The GenoDiabMar registry has collaborated on clinical, analytical and other biomarker data with other registries interested in diabetes, but the full description of this registry included in this manuscript has not been previously published.

We have tried our best to improve the manuscript according to your comments as well as our own consideration. We earnestly appreciate for your works, and hope that the revised manuscript will be reviewed and considered for publication in Journal of Clinical Medicine.

Once again, thank you very much for your comments and suggestions

Reviewer 2 Report

The revised manuscript contains substantial amendments in line with the reviewer comments. Overall, the revised manuscript is a major improvement over the originally submitted version. and seems suitable for publication in JCM a minor linguistic revision.

Author Response

Dear reviewer 
Thanks a lot for your comments. The manuscript has been reviewed by an english native colleague and we will re submit with minimal linguistic changes to the editorial team. 

Please find enclosed the final version
